# Magnetic field boosted ferroptosis-like cell death and responsive MRI using hybrid vesicles for cancer immunotherapy

Bo Yu[1], Bongseo Choi [1], Weiguo Li [1,2] & Dong-Hyun Kim [1,2,3,4 ✉]

We report a strategy to boost Fenton reaction triggered by an exogenous circularly polarized magnetic field (MF) to enhance ferroptosis-like cell-death mediated immune response, as well as endow a responsive MRI capability by using a hybrid core-shell vesicles (HCSVs). HCSVs are prepared by loading ascorbic acid (AA) in the core and poly(lactic-co-glycolic acid) shell incorporating iron oxide nanocubes (IONCs). MF triggers the release of AA, resulting in the increase of ferrous ions through the redox reaction between AA and IONCs. A significant tumor suppression is achieved by Fenton reaction-mediated ferroptosis-like cell-death. The oxidative stress induced by the Fenton reaction leads to the exposure of calreticulin on tumor cells, which leads to dendritic cells maturation and the infiltration of cytotoxic T lymphocytes in tumor. Furthermore, the depletion of ferric ions during treatment enables monitoring of the Fe reaction in MRI-R2* signal change. This strategy provides a perspective on ferroptosis-based immunotherapy.

[1] Department of Radiology, Feinberg School of Medicine, Northwestern University, Chicago, IL 60611, USA. [2] Department of Bioengineering, University of Illinois at Chicago, Chicago, IL 60607, USA. [3] Robert H. Lurie Comprehensive Cancer Center, Chicago, IL 60611, USA. [4] Department of Biomedical Engineering, McCormick School of Engineering, Evanston, IL 60208, USA. ✉email: dhkim@northwestern.edu

Chemotherapy and chemotherapy-based combination therapies have been developed and commonly performed for cancer treatment in clinics[1]. Due to the immune system contribution to the eradication of tumors, considerable attention has been given to developing chemotherapy-based immunotherapies[2–4]. Although the traditional strategy, apoptosis, is still widely used in chemotherapy for treating cancer, upcoming evidence shows its limitation for immunotherapy, such as inducing tumor resistance[5], weakening co-stimulation of T cells, and even secreting immunosuppressive cytokines[6]. Compared with apoptosis, ferroptosis programmed cell death mechanisms, that is not dependent on apoptosis, have attracted excellent cancer research attention[7,8]. In the past few years, ferroptosis has been developed as a form of regulated cell death since it was first identified in an experimental context by applying the chemical inhibitor Erastin on cancer cells in 2012[9,10]. So far, several signaling pathways of ferroptosis have been identified with cytological characteristics. The increase of intracellular Fe ions and follow-up Fenton reaction, which elevates reactive oxygen species (ROS) levels, lead to ferroptosis cell death by irresistible lipid peroxidation. The intense membrane lipid peroxidation and consequential loss of selective permeability of the plasma membrane are characterized in ferroptosis[11,12]. Another signaling pathway is involved with the inactivation of glutathione-dependent peroxidase 4, resulting in ferroptosis[12–14]. Generally, ferroptosis-induced cell death has been proved to be effective in killing cancer cells through ROS accumulation in cells[15,16]. Moreover, recent studies prove that the oxidative stress-inducing ferroptosis also upregulates the translocation of calreticulin (CRT) expression on the surface of tumor cells. The prophagocytic eat me CRT signal induces robust antitumor immune responses by eliciting phagocytosis of tumor-associated antigens[2,17,18]. The additional potential to trigger immune response is a promising feature of ferroptosis with the latest advances in immune cancer therapies.

Iron-based nanomaterials such as superparamagnetic iron oxide nanoparticles[19], iron nanometallic glasses[20], iron cross-linked gel nanoparticles[21], and the iron-based metal−organic frameworks[22] have been extensively tested as ferroptosis-inducing agents recently. Their capability of ROS generation through Fenton reaction of ferrous ($Fe^{2+}$) or ferric ($Fe^{3+}$) ions is believed to play a critical role in achieving a sufficient cancer therapeutic effect. Those iron-based nanomaterials improve the tumor specificity and therapeutic efficacy of ferroptosis cancer therapy to some degree. However, there are still some challenges, such as unwanted toxicity of carrier, low ROS conversion efficacy, required extra components for combinational effects, and a high dosage of nanomaterials that generally make it hard to be widely available.

Herein, we hypothesize that on-demand localization of Fenton reaction can significantly enhance both ferroptosis and the ferroptosis-like cell death mediated immune response triggered by an exogenous circularly polarized magnetic field (MF), as well as endow a responsive magnetic resonance imaging (MRI) capability by using a hybrid core–shell vesicles (HCSVs). HCSVs are synthesized to have an ascorbic acid (AA) core and iron oxide nanocubes (IONCs) embedded poly lactic-co-glycolic acid (PLGA) shell layer. The loaded AA in the core of HCSVs is used to reduce the ferric in the IONCs to ferrous after destroying the vesicle shell in the circularly polarized MF. On the other hand, the ratio change of ferric to ferrous leads to the responsive MRI. There are three important aspects reported in this therapy strategy. First, biosafe materials, AA, PLGA, and iron oxide IONCs are used to form the therapeutic carrier. Second, exogenous MF-triggered release of AA lead to the redox reaction of IONCs and enables ferroptosis-like cell death and the follow-up immune response. Third, MRI R2* signal is able to predict the Fenton reaction boosted by the exogenous MF.

## Results

### Exogenous MF-boosted ferroptosis-like cell death triggered calreticulin exposure via Fenton reaction by using HCSVs.
As shown in Fig. 1a, once AA is released under the MF, ferric ions in IONCs are reduced by AA increasing ferrous ions, which are a stronger inducer for Fenton reaction compared with ferric ions[23]. The enhanced ferroptosis efficiently inhibits tumor growth with an enhanced immune response. MRI is able to monitor the ratio change between ferrous and ferric ions. It demonstrates that our on-demand MF-boosted Fenton reaction enhances the ferroptosis-like death mediated cancer therapeutic efficacy for the treatment of TRAMP-C1. This strategy may provide a perspective on the use of ferroptosis-like cell death based cancer therapy.

AA hydrophilic core and IONCs-decorated shell structure of HCSVs were confirmed by the TEM images (Fig. 1b). The obtained HCSVs were easily collected by the magnetic bar as shown in Supplementary Fig. 1, and the collection procedure did not influence the HCSVs. The AA weight ratio in HCSVs was found to be 31.2% (Supplementary Fig. 2). External circularly polarized MF application remotely deformed the shape of HSCVs. During MF application, the embedded IONCs in the shell of HCSV started to split off from the HCSVs in 5 min. Then, 10 or 20 min of MF treatment efficiently disrupted the IONCs embedded shell of HCSVs resulting in IONCs in the shell budded off from the HCSVs, as shown in Fig. 1b. Following further 6 h incubation, the 10 min MF treatment group demonstrated the cube-like empty structure, which signified the leaked AA from the core of HCSVs and dissolved the IONCs. Thus, while a static magnetic field did not affect the shape of HCSVs, we speculated the circularly polarized magnetic field forced the circular back-and-forth movement of the IONC embedded shell, resulting in deformed shell layers and AA release. Subsequently, the released AA in the core could be utilized for the redox reaction of ferric ions on the IONCs to be ferrous. Ascorbic acid (AA, $C_6H_8O_6$) can act as an electron-donor reducing ferric ions to ferrous state and finally forms dehydroascorbic acid[24,25]. As shown in Fig. 1c, Supplementary Figs. 3, 4, after 10-min MF treatment, the increase of ferrous ions was found in 0.1 h. The concentration of ferrous ions continued to increase when we prolonged the incubation time. In contrast, the HCSVs without MF treatment showed no changes of ferrous ion concentration in the same 6 h incubation period. The decrease absorbance of the mixture of ferric ions was found in Supplementary Fig. 5, which matched with a tendency of increase of ferrous ions. This result coincided with the shape of MF treated HCSVs in TEM images. Furthermore, the external MF-triggered release of core materials was confirmed with a co-incubation of cancer cells and HCSVs loaded with doxorubicin hydrochloride (Dox) instead of AA. We found the external MF treatment triggered the release of Dox from HCSVs and covered the whole cells, including cell nucleus, in confocal images (Supplementary Fig. 6). In contrast, non-MF treated group showed intact and steady red fluorescent HCSVs surrounding the tumor cells. Interestingly, when the PLGA concentration was doubled, thicker shelled HCSVs were obtained (Supplementary Figs. 7a and 8, There is a single peak with the size range 300 nm.). However, HCSVs with a thick shell did not show the response to the MF treatment. While increasing PLGA concentration, the solid HCSVs not affecting external MF treatment were formed (Supplementary Fig. 7b). Thus, optimizing the shell thickness in HCSVs was critical to endow the MF responsiveness to stimulate the increase of ferrous by the redox reaction.

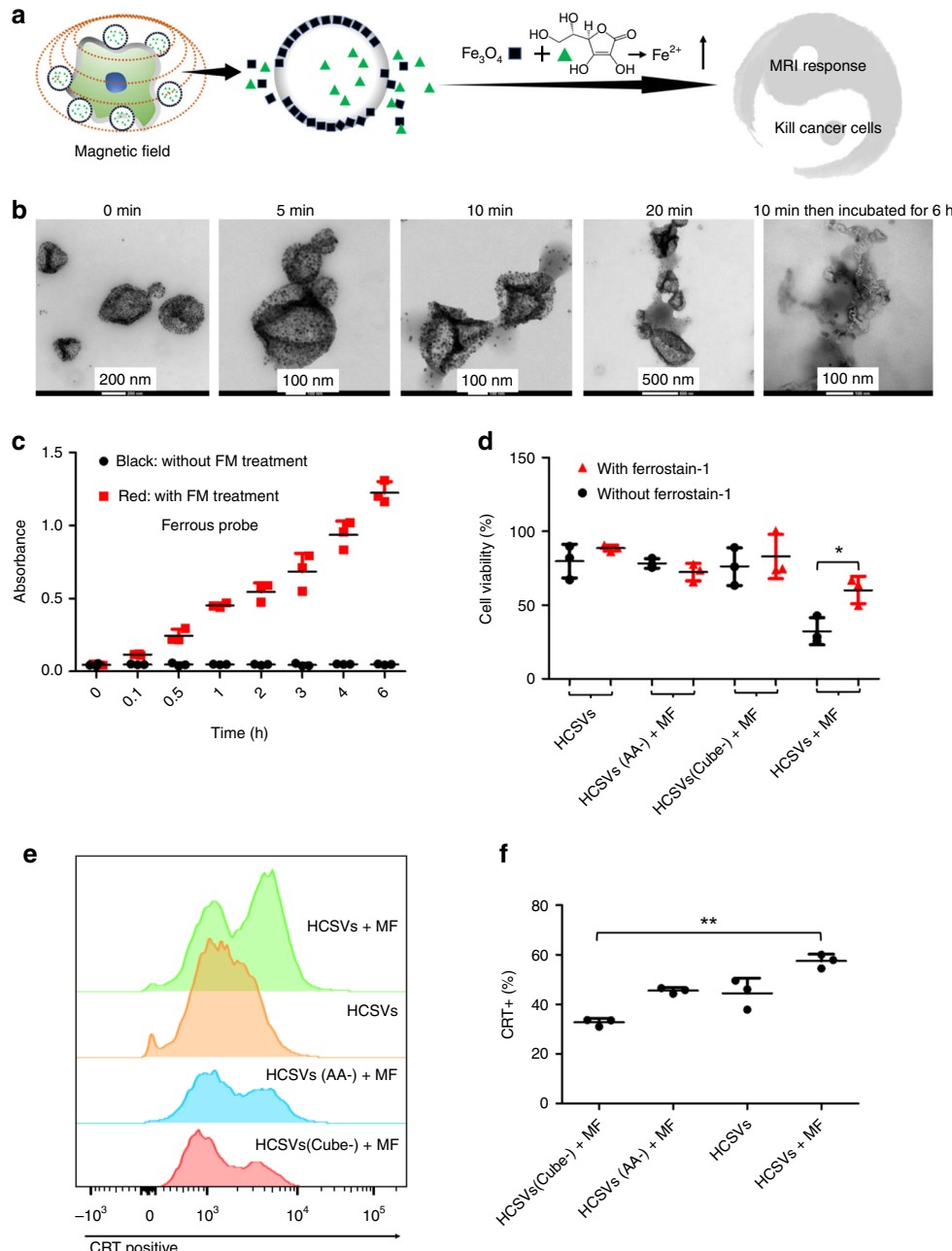

**Fig. 1 Exogenous MF-boosted Ferroptosis-like cell death. a** Scheme of exogenous MF-boosted Fenton reaction and responsive magnetic resonance imaging (MRI) using HCSVs, **b** TEM images of HCSVs treated with the circularly polarized magnetic field (MF, 2 Hz) for various time-periods. Experiments were repeated three times. **c** Relative concentration change of Ferrous ions tested by the ferrous probe, $n = 3$ independent samples. Data are shown as means ± s.d. **d** Ferrostatin-1 rescued TRAMP-C1 cells treated with HCSVs with/without MF, $n = 3$ independent samples. $P = 0.017$ **e** Flow cytometry assay of calreticulin (CRT) exposure on the surface of TRAMP-C1 after various treatment. **f** Relative quantification of Flow cytometry assay of CRT exposure. $n = 3$ independent samples. Data are shown as means ± s.d. $p < 0.05$, two-tailed paired $t$-test. Source data are provided as a Source data file.

The generated ferrous ions, inducing stronger Fenton-like oxidation than ferric ions, triggered the higher accumulation of ROS, and finally inhibited tumor cell growth (Fig. 1d and Supplementary Fig. 9). The dichlorofluorescein diacetate staining assay was utilized to determine whether the combination of MF and HCSVs induced strong ROS generation. As shown in Supplementary Fig. 10, HCSVs and MF treated groups demonstrated significant ROS accumulation in the cells. When IONCs or AA was excluded in HCSVs (HCSVs (Cubes-) or (HCSVs (AA-)), there was no obvious ROS accumulation after being treated with MF. The ROS level in the only HCSVs group without

MF was also similarly low with the previous two groups, which collectively were still much weaker than the combination group (HCSVs plus MF). Although AA is considered to be the scavenger of cytoplasmic ROS and an inhibitor of ROS induced apoptosis, AA shows a poor ability in rescuing ferroptosis cells[22]. Herein, the AA release from HCSVs almost occurred outside the cells, since most of the HCSVs just attached to the cell membranes after 6 h coincubation. Based on the results, we concluded that ferroptosis was mainly induced by the combination of MF and HCSVs. Ferrostatin-1, a small-molecule inhibitor of ferroptosis was used to confirm the induced ferroptosis by our HCSVs and

MF. As shown in Fig. 1d, when ferroptosis is inhibited by Ferrostatin-1 in the HCSVs and MF treated group, the cell viability was significantly enhanced. Furthermore, the HCSVs and MF treated group demonstrated the upregulation of lipid peroxidation (LPO), one of markers for oxidative stress involving ferroptosis (Supplementary Figs. 11, 12). C11-BODIPY581/59, an oxidation-sensitive fluorescent lipid peroxidation probe, have been widely used as a detector for ferroptosis[26]. The shift of red to green fluorescence indicated the lipid peroxidation. As the transferrin receptor on the cell membranes will affect the cellular uptake of ferrous and ferric ions[27], herein, mesoporous silica nanoparticles (MSN) was used as a carrier for both ferrous and ferric ions. As shown in Supplementary Fig. 11, cells treated with ferrous-MSN showed a higher green fluorescence as compared with ferric-MSN treated cells which indicated higher level of LPO. This might be due to a stronger Fenton reaction induced by ferrous ions as compared with ferric ions[23]. A higher ratio of green to red signal also found in the combination of HCSVs and MF.

There are two main pathways, which are iron metabolism and reactive oxygen species metabolism, respectively, involved in ferroptosis[28]. Herein, our MF controllable therapeutic system, HCSVs, was directly used to supply ferrous ions to cancer cells. This strategy was directly related to boost iron metabolism, which led to radical accumulation and result in lipid hydroperoxides. For the ROS metabolism, accumulated lipid hydroperoxides in cells are normally detoxified by a GPX family member called glutathione peroxidase 4 (GPX4) that uses glutathione to convert lipid hydroperoxides to lipid alcohols and thus represses ferroptosis[27]. As shown in Supplementary Fig. 13, the combination treatment of MF and HCSVs downregulated GPX4. On the other hand, Ferrostatin-1 treatment recovered GPX4 level inside cells. We assumed that the continuous depletion of GPX4 by HCSVs-boosted lipid hydroperoxides contributed to the downregulation of GPX4, while the protection of Ferrostatin-1 reduced the depletion of GPX4 that finally increased the GPX4 level inside cells. Taken together, the combination of HCSVs and MF efficiently inhibited TRAMP-C1 growth through ferroptosis-mediated cell death.

Oxidative stress has shown the ability to induce upregulation of CRT expression (prophagocytic eat me signal) as well as the induction of cell death. The effect of triggered oxidative stress from the combination HCSVs and MF on CRT expression was determined by flow cytometry of TRAMP-C1 cells treated with HCSVs and MF (Fig. 1e, f). At 6 h post-treatment of TRAMP-C1 cells, the combination of HCSVs plus MF ($57.5 \pm 2.7$) significantly promoted CRT exposure. In contrast, HCSVs (AA-) plus MF ($45.6 \pm 1.2\%$) and only HCSVs ($44.5 \pm 6\%$) moderately induced CRT exposure on the surface of TRAMP-C1 comparing to HCSVs (Cube-) plus MF ($33 \pm 1.2\%$). It is noteworthy to mention that the boosted ferroptosis and CRT accumulation followed by HCSVs and MF treatment will be promising for combinational immune cancer therapy applications.

**Responsive MRI monitored exogenous MF-boosted Fenton reaction.** Ferrous and ferric iron ions show significantly different relaxation behaviors in MRI. It was reported that the R2* signal of ferrous is weaker than that of ferric[29]. Herein, MRI was further utilized to monitor the ferrous and ferric ions conversion during the HCSVs and MF treatment. As shown in Fig. 2a, the R2* values of IONCs (AA+) treated with MF were relatively constant in a concentration range of 0–30 μg/ml of IONCs while IONCs without AA samples showed proportionally increasing R2* values with the concentration of IONCs. We speculated MRI R2* scans could monitor the MF triggered ferric/ferrous conversion with

HCSVs. To confirm the hypothesis, in vivo MRI R2* scans were performed in mice tumor models treated with I.T. injection of HCSVs only or I.T. injection of HCSVs plus MF. As shown in Fig. 2b, c, the HCSVs and MF treatment led to a significantly higher R2* signal compared with HCSVs alone group. The depletion of ferric ions during treatment contributed to R2* signal decrement-enabled monitoring the reaction by MRI. Thus, MRI R2* change boosted by the external MF enabled us to predict the Fenton reaction.

**Ferroptosis-like cell death mediated anticancer effect.** Subsequently, the in vivo anticancer effect of combination HCSVs and MF was demonstrated in TRAMP-C1 bearing mice model as a proof-of-principle trial. Four different groups of HCSVs, HCSVs (IONCs-) + MF, HCSVs (AA−) + MF, and HCSVs + MF were utilized to prove the anticancer effect of HCSVs and MF treatment. As shown in Fig. 3a, the treatments were performed every 3 days. The results from the tumor growth curves (Fig. 3b) were in line with the result of final tumor weight, as shown in Fig. 3d. The mouse weight indicated that all treatment strategies did not show significant toxicity to the tested animals (Fig. 3c). In Fig. 3d, e, a group treated HCSVs + MF showed superior tumor growth inhibition among experimental groups at 14 days post-treatment. Other control groups that treated only HCSVs, HCSVs (AA-) + MF and HCSVs (Cube-) + MF showed moderate tumor growth inhibition efficacy compared with the HCSVs + MF treated group. The average weight of tumor in each group were found to be $3.1 \pm 0.2$, $2.1 \pm 0.8$, $1.4 \pm 0.3$, and $0.4 \pm 0.3$ g after treating with HCSVs (Cube-) plus MF, HCSVs (AA-) plus MF, HCSVs and HCSVs + MF, respectively. Terminal deoxynucleotidyl transferase dUTP nick end labeling (TUNEL)-stained tumor slices, which were collected 24 h post 1st local treatment, were shown in Fig. 3f. The TUNEL stain showed a robust conclusion on the therapeutic effect of ferroptosis. At 14-day post treatment, the major organs (i.e., liver, spleen, lung, and kidneys), which might be affected by nanomaterials-based therapy[30], were sliced and stained with Hematoxylin and Eosin (H&E) for histology analysis. The results demonstrated that there is no noticeable tissue damage in the collected organs (Supplementary Fig. 14). In addition, we tried to treat the animal with the combination of MF and HCSVs through i.v. of HCSVs. A dramatic growth curve was found and there was no sign of therapeutic effect was proved (Supplementary Fig. 15). This might be due to low tumor accumulation efficiency of HCSVs, which is the main challenge in nanomedicine[31]. The scope in work was to prove an MRI visible and MF controllable method by utilizing HCSVs with a special core–shell design through I.T., which a local therapy modality was developed to treat cancer as the clinical method in Radiology[32,33]. Collectively, it was proved that the exogenous magnetic field-boosted Fenton reaction efficiently inhibit tumor growth.

**Ferroptosis-like cell death mediated anticancer immune responses.** Encouraged by the efficiency of inhibiting in vivo tumor growth with as-demonstrated CRT upregulation and exposure after the treatment of HCSVs and MF, we further harvested the draining lymph node (LN) and tumor to examine how this combination therapy interacts with the immunological system. It is well known that released tumor-associated antigens such as CRT upregulation/exposure cells will be captured by antigen-presenting cells such as dendritic cells (DCs). The maturation of DCs then activates T cells to generate CD8+ T cells in lymph nodes. Finally, the generated cytotoxic T lymphocytes (CTLs, CD3+CD4−CD8+) migrate to tumor sites for killing tumor cells. Herein, a splenocytes proliferation assay was performed to confirm if CRT exposure contributed to increase

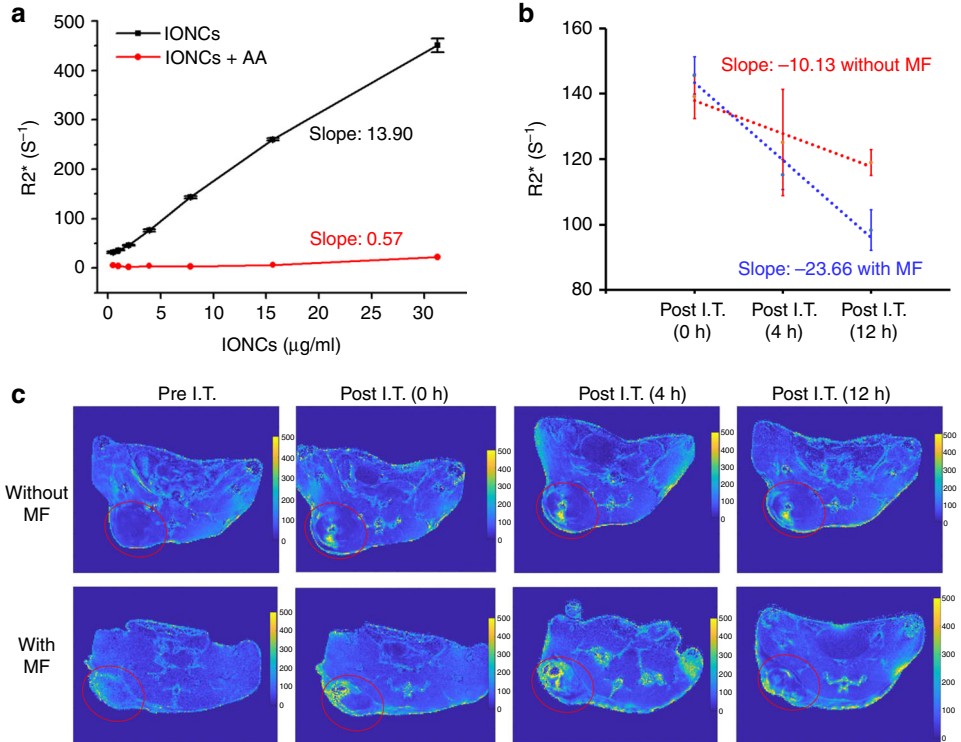

**Fig. 2 Responsive MRI. a** In vitro R2* value of IONCs treated with/without AA, R square are 0.9963 and 0.8287 for IONCs (black cure) and IONCs + AA (red curve), respectively. **b** In vivo R2* change after intra-tumoral injection of HCSVs treated with/without MF. R square are 0.9520 and 0.9741 for red curve (with MF) and blue curve (without MF), respectively. ($n = 3$ independent samples) Data are shown as means ± s.d. **c** Related R2* mapping after intra-tumoral injection of HCSVs treated with/without MF (red cycle indicated tumor region). Source data are provided as a Source data file.

immune response in the combination therapy. As shown in Supplementary Fig. 16, the combination group of HCSVs+MF remarkably promoted the proliferation of splenocytes by near two folds of control group. After blockade of the exposure CRT on the treated TRAMP-C1 cells by anti-CRT at 4 °C for 1 h, the optical density at 450 nm was significantly decreased to be $1.16 \pm 0.24$. Taken together, CRT exposure contributed to increase immune response in the combination therapy. Furthermore, the HCSVs and MF treatment-induced immunological responses including in vivo DC activation in inguinal LNs, and T cells activation both in LN and tumors, were examined at 8-day of post 3rd treatment. As shown in Fig. 4a, b, the combination treatment of HCSVs + MF significantly promoted the maturation of DCs ($35.88 \pm 1.8\%$ (including CD80+, CD86+, and CD80+ plus CD86+cell population)) compared with the control group treated with HCSVs (Cube-) + MF ($17 \pm 6.3\%$). Also, the maturation DCs in HCSVs (AA-) + MF treated group and HCSVs only group were decreased to be $25.5 \pm 4.4\%$ and $27.9 \pm 0.9\%$, respectively. The higher CRT upregulation/exposure on tumor cells treated by HCSVs + MF generate stronger DCs-based immune responses.

Furthermore, both T cells activation in LN and the migration of related CTLs to the tumor site play a vital role to achieve successful elimination of tumors. Therefore, the CTLs (CD3+CD4−CD8+ T cells) level in draining LNs was analyzed in all four groups. As shown in Fig. 4c, e, the group treated with HCSVs + MF demonstrated the highest population of CD8+ CTLs ($46.2 \pm 1.9\%$) in the draining LNs among the experimental groups of HCSVs (AA-) + MF ($42.5 \pm 3.1\%$) and HCSVs only group ($43.4 \pm 0.9\%$) and HCSVs (Cube-) + MF ($40.2 \pm 3.2\%$). This result indicates that the HCSVs + MF treatment-inducing ferroptosis and CRT exposure of cancer cells significantly increases the generation of CD8+ CTLs. Notably, CD8+ CTLs infiltration level

in the tumor for the group treated with HCSVs + MF was also found to be highest among all four groups (Fig. 4c, d). From left column to right column in Fig. 4d, the ratio of CD8+ CTLs was $3.5 \pm 0.9\%$, $5.0 \pm 1.7\%$, $4.3 \pm 0.5\%$, and $7.7 \pm 0.8\%$, respectively. Taking in vivo data together, the HCSVs + MF treatment demonstrated significant inhibition of TRAMP-C1 tumor growth by ferroptosis and CRT mediated CD8 + CTLs infiltration in the tumor.

## Discussion
An MF-responsive theranostic HCSV platform composed by AA loaded core and IONC embedded shell was prepared by controlling the shell thickness of HCSVs. Exogenous circularly polarized MF application to HCSVs-triggered Fenton reaction reducing the ferric of IONC to be ferrous with released AA from the core of HCSVs. The enhanced oxidative stress by the MF application to HCSVs was utilized to induce ferroptosis-mediated anticancer effect. As the ratio change of ferric to ferrous resulted in R2* decrease, the MRI monitoring of Fenton reaction was enabled as well as MRI tracking of HCSVs. Finally, the HCSVs and MF treatment demonstrated significant inhibition of tumor growth in TRAMP-C1 tumor-bearing mice model. Besides the ferroptosis cell death by the treatment, the oxidative stress caused by Fenton reaction also induced the translocation of CRT on the surface of tumor cells. The CRT exposure led to dendritic cells maturation in draining lymph nodes and enhanced cytotoxic T lymphocyte infiltration to tumor sites. The CRT exposure eventually contributed to significant tumor growth inhibition of the HCSVs and MF treatment. This HCSVs and MF treatment strategy provide a perspective for the use of ferroptosis-based cancer therapy.

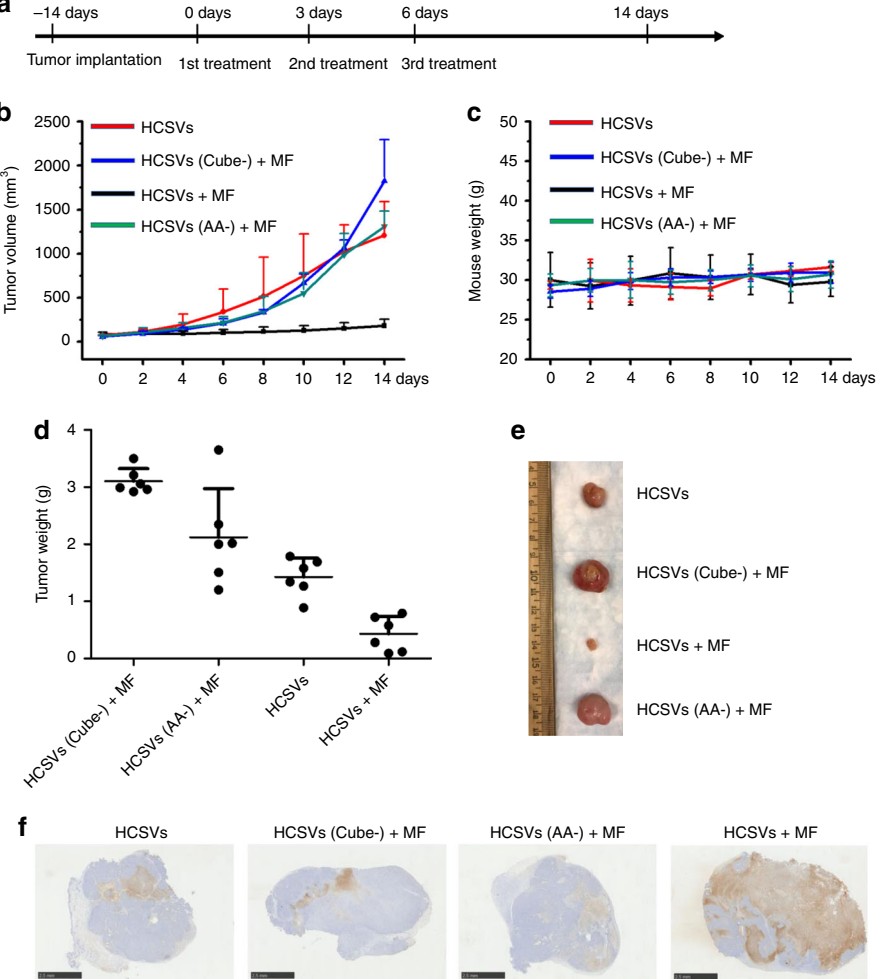

**Fig. 3 Combination therapy-mediated antitumor effect in a TRAMP-C1 tumor model. a** in vivo treatment timeline. **b** tumor growth curves of the TRAMP-C1. **c** the record of mouse weight. **d** tumor weight at the end time point. ($n = 6$ independent samples). Data are shown as means ± s.d. **e** the typical ex vivo photo of dissected tumors from various treatment. ($n = 6$ independent samples) **f** TUNEL-stained tumor slices (scale bar: 2.5 mm). Tumor tissues collected from different groups at 24 h after various treatments. Experiments were performed one time. Source data are provided as a Source data file.

## Methods

**Materials**. All reagents and solvents were obtained commercially and used without further purification. Iron (III) chloride hexahydrate (≥98%), n-docosane (99%), 1-octadecene (90%), n-hexane, acetonitrile, ethanol, dichloromethane ($CH_2Cl_2$), poly (d,l-lactide-co-glycolide) (lactide:glycolide 50:50, PLGA), and poly(vinyl alcohol) (Mw 89,000–98,000, 99+% hydrolyzed, PVA) were purchased from Sigma–Aldrich Co (Milwaukee, WI). Sodium oleate (>97%) was purchased from TCI America (Portland, OR). Iron oleate was prepared by following a reported procedure[34].

**Synthesis of iron oxide nanocubes (IONC)**. IONC were synthesized by the heat-up method (thermal decomposition method)[34]. The 1.57 g of the prepared iron oleate, 0.53 g of sodium oleate, and 6.0 g of n-docosane were mixed with 11 mL of 1-octadecene in a 150 mL three-neck round-bottom flask. Under vacuum, the reaction mixture was stirred for 1 h at 120 °C. Then, the reaction mixture was heated to 337 °C with a heating rate of 3–5 °C/min under nitrogen flow protection and maintained at 337 °C for 30 min. Then removing the heating system, the reaction mixture was cooled to 80 °C. A mixture of n-hexane (20 mL) and ethanol (30 mL) was used to precipitate final product. The precipitates were collected with centrifuge at 14,000 × $g$ for 30 min. The washing process was repeated three times. Finally, the IONC were dried under vacuum and redispersed in n-hexane.

**Preparation of the hybrid core-shell vesicles (HCSVs)**. HCSVs were prepared through a double emulsion (water/oil/water) method. First, water-in-oil emulsion was prepared by adding 1 mL of AA water solution into 5 mL of dichloromethane solution (PLGA is 5 mg/mL, IONC is 0.2 mg/mL) in the ice-water sonication bath for 0.5 h. To prepare the second emulsion, the water-in-oil emulsion was added into 30 mL of PVA water solution (2 mg/mL) using a vortex for 5 min and

following the treatment of a homogenizer in ice bath for 10 min to form w/o/w HCSVs. The PLGA sediment was removed by 2500 × $g$ centrifugation for 10 min. The products were purified by centrifugation (1200 × $g$, 1 min) to remove micro sized PLGA (the precipitate). Subsequently, HCSVs in the supernatants were collected by 14,000 × $g$ centrifugation for 5 min and were washed by the ultrapure water. To prepare HCSVs-Dox, AA water solution was replaced by the same concentration of Dox. HCSVs (Cube-) were prepared by following the same procedure without adding IONC. HCSVs (AA-) were prepared by following the same procedure without adding AA. Size distribution was analyzed using a Zetasizer Nano ZSP (Malvern Instruments, Malvern, UK).

Synthesis of MSN and iron ions loaded MSN. MSN were synthesized by an oil–water biphase reaction approach by following a reported procedure[34]. In details, 20 mL of CTAC solution (25 wt%) and 0.01 g of TEA were mixed with 30 mL of water and gently stirred at 50 °C for 1 h. Then 15 mL of TEOS in cyclohexane (5% v/v) was slowly added to the above solution and kept at 50 °C for another 18 h. Finally, MSN were collected by centrifugation at 14,000 $g$ for 15 min. The precipitates were washed four times (24 h/time) with 1% (wt%) NaCl/ methanol solution to remove CTAC. The structure of MSN was confirmed by TEM. Iron ions was dissolved in the water. The amount of water is 1.5 times the weight of MSN. Then the dry MSN powder was mixed with iron ions aqueous solution for 2 h. Above mixture of MSN and iron ions was dry under Argon flow overnight. Finally, iron ions loaded MSN was washed with ethanol and was kept in dry powder for next application. The iron weight ratio in MSN were 17.9% and 18.4% for ferrous and ferric ion, respectively.

**Iron ions concentration**. After digested by the mixture of $HNO_3$ (67%) and $H_2O_2$ (30%), the concentration of Fe was measured by ICP-MS. The iron weight ratio in

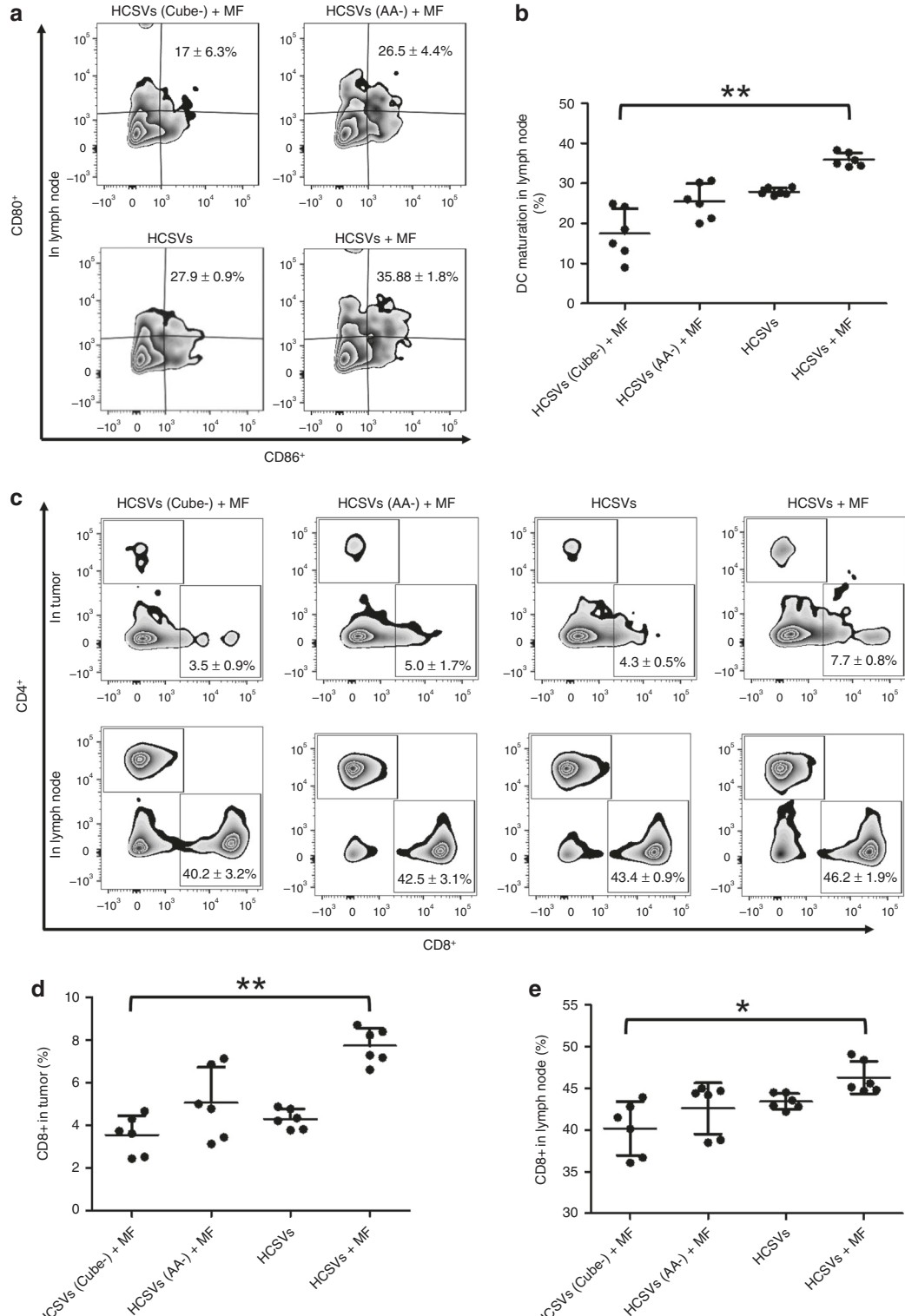

**Fig. 4 Immune responses after treatment to TRAMP-C1 tumor-bearing mice.** Flow cytometric analysis images (**a**) and the statistical data (**b**) for in vivo DC maturation. Cells in the tumor-draining lymph nodes were collected after various treatments for the assessment by flow cytometry after staining with CD11c, CD80, and CD86. **$p = 0.0044$. Representative flow cytometric analysis images showing $CD8^+$ T cells ($CD3^+$ $CD4^-$ $CD8^+$) from different groups of mice. Proportions of tumor infiltrating $CD8^+$ killer T cells in the tumor (up line) and draining lymph nodes (down line) (**c**) and the statistical data in the tumor (**d**) (**$p = 0.0013$) and in lymph node (**e**) among all cancer cells (*$p = 0.0491$) ($n = 6$ independent samples). The percentage values in the graphs were defined in relation to T cell populations. Data are shown as means ± s.d. two-tailed paired $t$-test. Source data are provided as a Source data file.

HCSVs was found to be 20.3%. And the iron weight ratio in HCSVs (AA-) was found to be 19.1%.

TEM Characterizations. Transmission electron microscopy (TEM) images were obtained on a FEI T12 microscope operated at an accelerating voltage of 120 kV. Standard TEM samples were prepared by dropping diluted products onto carbon-coated copper grids.

The loading efficiency of AA in HCSVs. 200 mg of HCSVs were mixed with 1 mL of dichloromethane and 0.1 mL water. The mixture was kept in centrifuge at $180 \times g$ for 1 h. The 100 μL of supernatant was collected and treated with 0.5 mM FeCl$_3$ in HCl solution (0.1 M) for 1 h. The absorbance is recorded at 260 nm. The AA weight ratio in HCSVs was found to be 31.2%. The molar ratio of iron ion to AA in HCSVs was ~1–2. And the AA weight ratio in HCSVs (Cube-) was found to be 27.3%.

The ferrous concentration. The ferrous concentrations were tested by using ferrozine (monosodium salt hydrate of 3-(2-pyridyl)-5,6-diphenyl-1,2,4-triazine-p, p′-disulfonic acid) which reacts with divalent Fe to form a stable magenta complex species is used. The maximum absorbance is recorded at 562 nm. After HCSVs were treated with the circularly polarized magnetic field, 0.5 mL of sample were extracted. Then the supernatant was collected by centrifuge at $17,500 \times g$ for 5 min, then was tested by ferrozine.

Confocal images of HCSVs-Dox treated cells. Briefly, the TRAMP-C1 cells were cultured on cover glass in six-well plates till 70% confluence were stained with DAPI for 20 min. After washing with PBS several times, the cells were incubated with HCSVs-Dox for 2 h, then treated with/without MF for 10 min. And the cells were continued to incubate for another 6 h. Finally, the cells were washed with PBS for three times then were fixed by 4% paraformaldehyde and visualized with a Nikon A1RS confocal microscope.

**Cell viability assay.** TRAMP-C1 cells ($1 \times 10^4$) were seeded in 96-well plates. And 18–24 h later, the cells were treated with or without MF for 10 min after incubation with various materials (HCSVs (Cube-), HCSVs (AA-), HCSVs) for 2 h. Then, all samples were incubated in the dark for another 24 h. At designed time point, 30 μL of MTT (5 mg/mL in PBS) was added into each well and follow-up incubation was performed for another 4 h. After removing the culture medium, the purple for-mazan was dissolved in 150 μL of DMSO. Finally, the absorbance of each well was recorded by a microplate reader at 560 nm. To confirm if the ferrostatin-1 could save the cells, 200 nM of ferrostatin-1 was added into related wells before MF treatment, and other procedure was kept the same.

**Cellular reactive oxygen species test by using DCFDA.** A final working concentration of 2 μM DCFDA was added and incubated for 40 min. Then the cells were washed by prewarmed PBS. Then the cells were treated with or without MF after incubation with various materials (HCSVs (Cube-), HCSVs (AA-), HCSVs) for 2 h. Finally, the cells were checked with a Nikon A1RS confocal microscope.

**Lipid peroxide (LPO) level measurement.** The freeze-thaw procedure was used to lyse the treated or untreated cells first. Then, the supernatants were collected from lysates by centrifuging at $12,000\, g$ for 15 min at 4 °C The supernatants were used for the MDA assay as described in the Lipid Peroxidation MDA assay kit (Sigma-Aldrich). LPO was calculated as nanomoles of MDA per milligram of protein. Lipid peroxide assay by C11-BODIPY581/591. The cells were stained with C11- BOD-IPY581/591 (2 μM) and incubated for 30 min. The cells were treated with various materials by following the procedure described in cell viability test. A microplate detection Cytation 3 (BioTek Instruments, Inc.) was used to detect the fluorescence intensity at 530 nm (green) and 591 nm (red).

**Western blot analysis.** TRAMP-C1 cells were seeded in the 8 cm culture dishes and cultured at 5% CO$_2$, 37 °C overnight. The cells were treated with various procedures, 400 μg/mL of HCSVs + MF + 200 nM of Ferrostain-1, 400 μg/mL of HCSVs, 400 μg/mL of HCSVs +MF, 600 μg/mL of HCSVs + MF, 800 μg/mL of HCSVs +MF, respectively, for 12 h. No treatment of TRAMP-C1 was used as control. The cell lysates were collected and analyzed by Western blotting according to the manufacturer's instructions. Antibodies were used listed as below: GPX4 Rabbit anti-Human, Mouse, Rat, Polyclonal, Invitrogen™, catalog number: PIPA579321, dilution 0.1–0.5 μg/mL; Goat Anti-Rabbit IgG H&L (HRP), Abcam, catalog number: ab205718, dilution 1:2000; β-Actin (8H10D10) Mouse mAb (HRP Conjugate), cell signaling, catalog number:12262, dilution 1:1000.

**MRI.** MRI studies were performed using a Bruker 7.0 T ClinScan high-field small animal MRI system with a commercial rat coil (Bruker Biospin). In vitro R2 star was performed after coincubation of IONC with/without AA at room temperature for 4 h (The molar ratio of Fe to AA approximately is 1:2). During in vivo test, body temperature was monitored continuously and controlled with a waterbed (SA Instruments, Stony Brook, NY). T2-weighted images were collected pre- and post-arterial infusion of the HCSVs (50 μL, 10 mg/mL). MR scans were performed using a gradient-echo sequence with following parameters: TR/TE = 1,300/7.2 ms, 1 mm slice thickness, FOV 71 × 85 mm, 216 × 256 matrix, respiratory triggering with

MRI-compatible small animal gating system (Model 1025, SA Instruments, Stony Brook, NY).

**In vivo therapy.** C57BL/6 mice from Charles River Laboratories, 8 weeks of age, Male. The animals were hosted in equipped animal facility with temperature at 68–79°F and humidity at 30–70%, under the same dark/light cycle (12:12). All animal studies were performed in accordance with protocols approved by the Institutional Animal Care and Use Committee Office at Northwestern University. For in vivo therapy, TRAMP-C1 tumor-bearing C57BL/6 mice were randomly divided into four treatment groups: the first group was intratumor injected (I.T.) with HCSVs saline solution (50 μL, 10 mg/Kg); the second group was intratumor injected with HCSVs saline solution (50 μL, 10 mg/kg) and follow-up treatment of MF for 10 min at 4 h post I.T.; the third group was intratumor injected with HCSVs (AA-) saline solution (50 μL, 10 mg/kg) and follow-up treatment of MF for 10 min at 4 h post I.T.; the fourth group was intratumor injected with HCSVs (Cube-) saline solution (50 μL, 10 mg/kg) and follow-up treatment of MF for 10 min at 4 h post I.T. ($n = 6$) Tumor growth curves of the TRAMP-C1 bearing mice treated with intra-venous injection (I.V.) of HCSVs (50 μL, 10 mg/kg) following with MF treatment 14 post I.V. ($n = 5$). And those mice were repeatedly treated three times at 0-day, 3-days, and 6-days time point. Finally, the tumor sizes were measured by a caliper every other day and calculated as volume = (tumor length) × (tumor width)$^2$/2.

**Histology.** Terminal deoxynucleotidyl transferase dUTP nick end labeling (TUNEL) stain. To evaluate the antitumor therapeutic efficacy, the tumor tissues were har-vested 24 h after various treatment described in vivo therapy. Hematoxylin and Eosin (H&E) stain. To analyze the toxicity toward the normal organs, liver, spleen, lung, and kidney were collected 14 days post treatment. The harvested tissues were fixed with 10% neutral formalin solution and then submitted to Mouse Histology and Phenotyping Laboratory core for TUNEL staining with 5 μm slice thickness.

**Flow cytometric analysis of calreticulin exposure on the cell surface.** TRAMP-C1 cells were seeded into the 6-well plate ($2 \times 10^5$ cells/well). And, ~18 h later, the cells underwent various treatment (HCSVs (Cube-) plus MF, HCSVs (AA-) plus MF, HCSVs only, and HCSVs plus MF) and cells were continued to culture for another 6 h. Then, the cells were harvested, washed twice with PBS, fixed in 0.25% paraformaldehyde for 5 min and incubated with purified mouse anti-calreticulin antibody (BD Pharmingen™, catalog number: 612136) for 30 min. After washed with PBS for three times, the cells were incubated with FITC-Goat anti-mouse secondary antibody (BD Pharmingen™, catalog number: 54001) and finally were examined by using BD LSRFortessa 6-Laser.

**Splenocytes proliferation assay.** Splenocytes obtained from treated mice were re-stimulated by antigens, and the cell proliferation was measured by a CCK-8 kit assay following previous report[35]. TRAMP-C1 cells in 6-well plate was treated with combination procedure of MF and HCSVs described in cell viability assay for 4 h, then was collected with trypsin. Then splenocytes ($5 \times 10^5$ per well) were seeded in the 96-well plate and retreated with the treated TRAMP-C1 cells ($5 \times 10^4$ per well) for 48 h. Splenocytes proliferation was measured using CCK-8 kit analysis. OD values at 450 nm were recorded by a microplate reader.

**Immune response analysis in vivo.** To examine DC maturation in vivo, TRAMP-C1 tumor-bearing C57BL/6 mice were treated with various strategy as described in in vivo therapy section. ($n = 6$) The inguinal lymph nodes were harvested 8 days post third treatment. The frequency of DC maturation in the LNs was then examined by flow cytometry after immunofluorescence staining with anti-CD11c-APC, anti-CD80-FITC, and anti-CD86-PE antibodies (BD Pharmingen™). To determine the intra-tumoral infiltration of CD8$^+$(CD3$^+$CD4$^-$CD8$^+$), lymphocytes were stained with anti-CD3-Brilliant violet 510, anti-CD4-PE, anti-CD8-PE-cy7. All antibodies were diluted with the ratio of 1:100. And, signal color beads (OneComp eBeads™ Compensation Beads) and unstained bead were used for compensation. The frequency of CD8$^+$(CD3$^+$CD4$^-$CD8$^+$) lymphocytes in the LNs were also exam-ined. Gating strategies used for cell sorting was shown in Supplementary Fig. 17.

**Data analysis.** Flow cytometry data was analyzed using Flow Jo V10. Microscopy images were analyzed using AxioVision Microscope Software 4.8 and ImageJ 1.52a. Data was presented as mean ± SD and analyzed in R Studio (statistical software). Student's t-test was used to assess differences between means. $P < 0.05$ was con-sidered statistically significant.

**Reporting summary.** Further information on research design is available in the Nature Research Reporting Summary linked to this article.

## Data availability
The source data underlying all quantitative figures are provided as a Source Data file. All data supporting the findings of this study is available from the corresponding author upon reasonable request. Source data are provided with this paper.

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

## Acknowledgements

This work was mainly supported by grants R01CA218659 and R01EB026207 from the National Cancer Institute and National Institute of Biomedical Imaging and Bioengineering. This work was also supported by the Center for Translational Imaging and Mouse Histology and Phenotyping Laboratory at Northwestern University.

## Author contributions

D.-H.K. and B.Y. proposed the research direction and guided the project. B.Y. and D.-H.K. designed and conducted the experiments, as well as analyzed the experimental results and drafted the manuscript. W.G.L. performed the MRI. B.Y. performed in vitro study and preparation of the materials. B.Y. and B.S.C. drew the pictures of schematic illustration. B.Y. performed in vivo animal experiments. D.-H.K. supervised the study, provided input for all of the experiments and the study concept, and edited the paper. All the authors checked the manuscript.

## Competing interests

The authors declare no competing interests.
