## [Peer Review File · Nature Communications]

Reviewers' comments:

Reviewer #1 (Remarks to the Author):

In this manuscript, the authors proposed a kind of hybrid core-shell vesicles (HCSVs) for cancer therapy capable of ferroptosis-mediated immune response boosted by exogenous magnetic-field and responsive magnetic resonance imaging. The HCSVs were synthesized to have an ascorbic acid (AA) core and iron oxide nanocubes (IONCs) embedded poly lactic-co-glycolic acid (PLGA) shell layer. All of the used materials are biocompatible. This is an interesting design for ferroptosis-based cancer therapy using biosafe materials. I recommend acceptance for publication after major revisions.

In this study, the loaded AA in the core of HCSVs was supposed to reduce the ferric in the IONCs to ferrous after destroying the vesicle shell in the circularly polarized magnetic field (MF). What's the product of AA after reducing ferric to ferrous? Data should be provided to demonstrate the product and thus reinforce the supposed redox reaction.

Relative concentration change of ferrous ions after the redox reaction was tested by ferrous probe. The concentration change of ferric ions after the redox reaction should also be measured to reinforce the supposed redox reaction.

Ferrous ion is a stronger inducer than ferric ion for Fenton reaction. This is an important hypothesis for the design of this study. However, no data were provided and only one literature was cited to demonstrate this point. Due to the importance, I think corresponding data should be supplemented.

The HCSVs in Supplementary Figure 6 looks bigger than that in Figure 1 b. So, the particle size should be not uniform. I am wondering whether there is one or two peaks in size distributions measured by DLS.

Due to the big particle size, I believe there are more HCSVs accumulated in liver and spleen than that in tumors. The authors should provide data of biodistribution in vivo to demonstrate the tumor accumulation of the HCSVs.

Data of HE staining should be presented to indicate non-toxicity of the HCSVs to normal tissues and toxicity to tumors.

Linear relationship should be presented in Figure 2 a, as well as the slope and R square.

Data were obtained at only two time points in Figure 2 b. It is too limited. At least 4 time points should be presented to draw a conclusion.

MR images of the tumor-bearing mice should be presented to show the MRI signal change at tumor site after injection of the HCSVs.

For the animal studies, intravenous injection should be used instead of intra-tumoral injection.

Reviewer #2 (Remarks to the Author):

The manuscript, “Exogenous Magnetic-field Boosted Ferroptosis-mediated Immune Response and Responsive Magnetic Resonance Imaging Using Hybrid Core-shell Vesicles for Cancer Therapy” investigates an approach to enhance the safety and efficacy of ferroptosis inducing nanoparticles as a cancer therapy. Using biosafe materials for iron oxide nanocubes, the authors synthesized hybrid core-shell vesicles (HCSVs). Notably, the Fenton reaction and subsequent ferroptosis induced by these nanoparticles can be triggered by an exogenous circularly polarized magnetic field (MF) – thus allowing for improved control over the release of the therapeutic effect. The authors show that the combination of HCSVs with MF results in strong suppression of tumor growth, however there are some concerns with the robustness of the in vivo experiments as well as the conclusions drawn from immunological mechanisms involved. Points to address include:

- The main concern with this manuscript is the claim that upregulation of calreticulin (CRT) is the key mechanism responsible for inducing/increasing immune response. It is shown to be correlated with immune cell activation but a causative effect is not validated. As such, it is recommended that an additional group is tested such that a blocking antibody against CRT is incorporated to verify that it is indeed a direct cause of improved DC and CTL responses reported.

Other concerns include:

- Clarification of number of mice used for each group in the in vivo experiments
- Clarification of timeline for tumor model – were TRAMP-C1 tumor cells and Treatment 1 both injected on same day? If so, treatment of mice with a more developed tumor would provide more confidence in the efficacy of the approach.
- The percentage values reported in the bar graphs of Figure 4 should specify whether they are defined in relation to all cells in tumor/lymph nodes or just relative to the T cell populations
- The flow cytometry graphs in Figure 4 should include percentages of the gated populations to corroborate the values shown in bar graphs.

Reviewer #3 (Remarks to the Author):

In the paper “Exogenous Magnetic-field Boosted Ferroptosis-mediated Immune Response and Responsive Magnetic Resonance Imaging Using Hybrid Core-shell Vesicles for Cancer Therapy” Yu et al present a Hybrid Core-shell vesicle (HCSV) vesicle able to release ascorbic acid (AA) in the presence of a magnetic field that subsequently leads to reduction of iron and its release. My comments to the paper restrain to the ferroptosis part as I cannot judge the novelty and critical aspect of the HCSV synthesis and usage.

Pointing this out there are several issues with the authors conclusion when they point out that ferroptosis is been triggered and this is the cause for the immune response observed; and additional experiments are required in order to substantiate their claims.

1) The authors claim ferroptosis is responsible for cell only based on a mild rescue effect observed with Ferrostatin 1. More evidence should be provided, do increase content of PUFA sensitize further? Does GPX4 OE expression is protective? This should be a minimal set to implicate lipid peroxidation in the phenomenon observed

2) The TRAMP-C1 cancer line deserve some characterization in the context of how it responds to standard ferroptosis inducing agents as well as providing expression data on key players such as GPX4, SLC7A11, γ -GCS, ACSL4 etc.

3) The in vivo effect is indeed dramatic and could well be that immune cells are further boosting the antitumor effect. If indeed ferroptosis is triggering this process, how ferroptosis inhibitors would modulate this response in vivo – Perhaps even more important would be to check how tumors respond in different genetic background such as Batf3 KO mice that fail to mount an innate immune response.

4) Methodologies used to monitor “ROS” and LPO are antiquated and several caveats for these methodologies have been extensively reported in the literature. More importantly the kit used to measure LPO is not measuring MDA as stated, without chromatographic separation this cannot be stated.

Response to the Referees' comments.

As we appreciate the reviewer's insightful critiques, we would gladly response to referees' comments as listed below. We also have revised our manuscript based on the review's comments.

Reviewer #1

General comments:

In this manuscript, the authors proposed a kind of hybrid core-shell vesicles (HCSVs) for cancer therapy capable of ferroptosis-mediated immune response boosted by exogenous magnetic-field and responsive magnetic resonance imaging. The HCSVs were synthesized to have an ascorbic acid (AA) core and iron oxide nanocubes (IONCs) embedded poly lactic-co-glycolic acid (PLGA) shell layer. All of the used materials are biocompatible. This is an interesting design for ferroptosis-based cancer therapy using biosafe materials. I recommend acceptance for publication after major revisions.

Response: *We really appreciate your all positive critics and comments. Please see our responses with additional data, and revised sentences marked with yellow background.*

Comment 1: In this study, the loaded AA in the core of HCSVs was supposed to reduce the ferric in the IONCs to ferrous after destroying the vesicle shell in the circularly polarized magnetic field (MF). What's the product of AA after reducing ferric to ferrous? Data should be provided to demonstrate the product and thus reinforce the supposed redox reaction.

Response: *We appreciate for the reviewer's comment. Ascorbic acid (AA; C₆H₈O₆) can act as an electron-donor reducing ferric ion to ferrous state and finally forms dehydroascorbic acid (Du, J. et al., Biochim Biophys Acta 2012, 1826, (2), 443-57.). In brief, AA is able to induce reductive dissolution of iron oxides to produce surface Fe(II), described by the following steps,*

Ascorbic Acid (C₆H₈O₆)

Dehydroascorbic acid (AAox)

*Upon AA reacted with iron oxide nanoparticles, ascorbic acid formed surface complexes (Fe(III)-AA) with the iron atoms on the surface of iron oxide (Fe(III)), followed by the electron transfer process from AA ligands to iron atoms, generating surface ferrous species (Fe(II)) and oxidized AA molecules (AAox, dehydroascorbic acid (DHA)). The Fe(II) will subsequently release dissolved ferrous ion (Fe²⁺) into the bulk solution phase slowly. Therefore, AA could increase the density of Fe(II). (Sun, H. et al., Applied Catalysis B: Environmental, 267 (15), 118383 (2020)). Comprehensive studies had been well established as discussed in above references. In **Figure 1c**, we have proved relative concentration change between ferrous and ferric ions tested by ferrous probe in the reaction system of AA and IONC (Fe₃O₄), which is a vital role to carry out ferroptosis. Additional sentence and reference were added to describe the AA reduction in the revised manuscript.*

(line 96 in Page 4)

Ascorbic acid (AA; C₆H₈O₆) can act as an electron-donor reducing ferric ions to ferrous state and finally forms dehydroascorbic acid (Du, J.; Cullen, J. J.; Buettner, G. R. Biochim Biophys Acta 2012, 1826, (2), 443-57.).

Comment 2: Relative concentration change of ferrous ions after the redox reaction was tested by ferrous probe. The concentration change of ferric ions after the redox reaction should also be measured to reinforce the supposed redox reaction.

Response: *We appreciate for the reviewer's comment. As reviewer mentioned, soluble ferric ions have strong absorbance at a broad range from 300 ~ 400 nm and can provide sufficient spectra to analyze its concentration. (Lohani, C. R., et al. Sensor Actuat B-Chem 2010, 143, 649) However, calculating absolute amount of ferric ion in the mixture of HCSVs and ions was challenging because of the overlapping absorbance of ferric ions and IONCs (250 ~ 400 nm) on HCSVs. (Yew, Y. P., et al. Nanoscale Res Lett 2016, 11.) Another*

challenge was that the absorbance of soluble ferrous ions was weak and not meaningful at the levels of ferrous iron typically present in groundwater. Thus, the absorbance at 320 nm which is minimizing the overlap of absorbance between IONCs and ferric ions was used to measure the relative changes of ferric ion reduction in the mixture. Herein, the change of relative absorbance at 320 nm after treating with or without magnetic fields was added as **Supplementary Figure 5**. After 10-min MF treatment, the decrease of total absorbance of reaction mixture at 320 nm was found in 0.1 h (**Supplementary Figure 5**). It indicates that the ferric ions are reacted with the released AA from the HCSVs core by the magnetic field. Finally, ferric ions in the IONCs form ferrous state. (Gupta, H., et al. *J Colloid Interface Sci* 2014, 430, 221) The decrease of absorbance continued to reduce while prolonging the incubation time. In contrast, the HCSVs without MF treatment showed no changes of relative absorbance. This result was line with the observation of the tendency of ferrous ions.

(line 101 in Page 4)

The decrease absorbance of mixture of ferric ions was found in **Supplementary Figure 5**, which matched with tendency of increase of ferrous ions.

Supplementary Figure 5. Relative absorbance change of the HCSVs treated with or without magnetic fields at 320 nm. Although soluble ferric ions have strong absorbance and provide sufficient spectra to analyze its concentration, calculating the ferric ions in the mixture of ions and HCSVs is still difficult because of the absorbance of IONCs on HCSVs. In addition, the absorbance of soluble ferrous ions is weak and not useful at the levels of ferrous iron typically present in groundwater. In contrast, ferric ions show a broad absorbance peak arranged from 300 nm to 400 nm. (Lohani, C. R., et al. *Sensor Actuat B-Chem* 2010, 143, 649) Thus, the change of relative absorbance at 320 nm could partially be used to measure the reduction of ferric ions in the mixture.

Comment 3: Ferrous ion is a stronger inducer than ferric ion for Fenton reaction. This is an important hypothesis for the design of this study. However, no data were provided and only one literature was cited to demonstrate this point. Due to the importance, I think corresponding data should be supplemented.

Response: *We appreciate for the reviewer's valuable comment. We fully agree with reviewer's comment. Ferrous ion is a stronger inducer than ferric ion. Here we tried to prove that using each ferrous or ferric ion loaded mesoporous silica nanoparticles (MSN) carriers and ferroptosis detector. C11-BODIPY581/59, an oxidation-sensitive fluorescent lipid peroxidation probe, have been widely used as a detector for ferroptosis (Martinez, A. M., et al. *Methods Mol Biol* 2020, 2108, 125). As there is the Transferrin receptor on the cell*

*membranes will affect the cellular uptake of ferrous and ferric ions, herein, mesoporous silica nanoparticles (MSN) was used as a carrier for both ferrous and ferric ions (100 µg/mL). Cells treated with ferrous-MSN showed a higher ferroptosis with enhanced green fluorescence as compared to ferric-MSN treated cells. This data is added in **Supplementary Figure 11**.*

(line 126 in Page 4)

C11-BODIPY581/59, an oxidation-sensitive fluorescent lipid peroxidation probe, have been widely used as a detector for ferroptosis.²⁶ The shift of red to green fluorescence indicated the lipid peroxidation. As the transferrin receptor on the cell membranes will affect the cellular uptake of ferrous and ferric ions,²⁷ herein, mesoporous silica nanoparticles (MSN) was used as a carrier for both ferrous and ferric ions. As shown in **Supplementary Figure 11**, cells treated with ferrous-MSN showed a higher green fluorescence as compared to ferric-MSN treated cells which indicated higher level of LPO. This might be due to a stronger Fenton reaction induced by ferrous ions as compared to ferric ions.²³ A higher ratio of green to red signal also found in the combination of HCSVs and MF

(line 263 in Page 9)

Synthesis of mesoporous silica nanoparticles (MSN) and iron ions loaded MSN. MSN were synthesized by an oil-water biphasic reaction approach by following a reported procedure.³⁴ In details, 20 mL of CTAC solution (25 wt %) and 0.01 g of TEA were mixed with 30 mL of water and gently stirred at 50 °C for 1 h. Then 15 mL of TEOS in cyclohexane (5% v/v) was slowly added to the above solution and kept at 50 °C for another 18 h. Finally, MSN were collected by centrifugation at 14000g for 15 min. The precipitates were washed four times (24 h/time) with 1% (wt %) NaCl/methanol solution to remove CTAC. The structure of MSN was confirmed by TEM. Iron ions was dissolved in the water. The amount of water is 1.5 times the weight of MSN. Then the dry MSN powder was mixed with iron ions aqueous solution for 2 h. Above mixture of MSN and iron ions was dry under Argon flow overnight. Finally, iron ions loaded MSN was washed with ethanol and was kept in dry powder for next application. The iron weight ratio in MSN were 17.9% and 18.4% for ferrous and ferric ion, respectively.

(line 308 in Page 10)

The cells were stained with C11- BODIPY581/591 (2 µM) and incubated for 30 min. The cells were treated with various materials by following the procedure described in cell viability test. A microplate detection Cytation 3 (BioTek Instruments, Inc.) was used to detect the fluorescence intensity at 530 nm (green) and 591 nm (red).

Supplementary Figure 11. Lipid peroxide stained with fluorescent C11-BODIPY581/591 in cells after coincubation with ferrous/ferric ions loaded mesoporous silica nanoparticles (MSN). (a) TEM images of MSN. (b) Fluorescence emission intensity of C11-BODIPY581/591 at 530 nm (green) and 591 nm (red) in cells after coincubated with ferrous-MSN (Ferrous-Green or Ferrous-Red) or ferric-MSN (Ferric-Green or Ferric-Red). (c) Ratio of green:red fluorescence intensity in cells after coincubated with ferrous-MSN or ferric-MSN. (d) Ratio of green:red fluorescence intensity in cells after coincubated with HCSVs following with/without MF treatment.

Comment 4: The HCSVs in Supplementary Figure 6 looks bigger than that in Figure 1 b. So, the particle size should be not uniform. I am wondering whether there is one or two peaks in size distributions measured by DLS. Due to the big particle size, I believe there are more HCSVs accumulated in liver and spleen than that in tumors. The authors should provide data of biodistribution in vivo to demonstrate the tumor accumulation of the HCSVs.

Response: *We appreciate for the reviewer's comment. To clarify the size distribution of HCSVs, the hydrodynamic size distribution was added in Supplementary Figure 8. There is a single peak with the size range of 300 nm. We agree that in vivo biodistribution of HCSVs will endow us a more sophisticated comprehension on the safety of nanoparticles. However, the scope of the current studies was to prove an MRI visible and MF controllable method by utilizing HCSVs with a special core-shell design through intra-tumor injection (I.T.) of therapeutic materials. Herein, an intra-tumoral local injection with MRI imaging was used to deliver the therapy agents. HCSVs had been efficiently given to the tumor through I.T. Furthermore, the relative concentration change of ferrous ions that is a key component in this design to trigger ferroptosis occurred in 6 hours (Figure 1c), which means the therapeutic effect of HCSVs was carried out in short time. Thus, during this period, HCSVs had been kept in tumor and other adverse toxic effects were not observed (Supplementary Figure 14). (Kamkaew, A., et al. Acs Appl Mater Inter 2016, 8, 26630.)*

(line 108 in Page 4)

Supplementary Figure 8, There is a single peak with the size range 300 nm.

(line 261 in Page 9)

Size distribution was analyzed using a Zetasizer Nano ZSP (Malvern Instruments, Malvern, UK).

Supplementary Figure 8. DLS analysis of hybrid core-shell vesicles with thin shell (red) and hybrid core-shell vesicles with thick shell (green). There is a single peak with the size range 300 nm.

Comment 5: Data of HE staining should be presented to indicate non-toxicity of the HCSVs to normal tissues and toxicity to tumors.

Response: *We appreciate the reviewer's insightful comment. As reviewer commented, to show the localized toxicity to tumors, TUNEL stained tumor tissues were added in Figure 3 and H&E stained organ tissues were added in Supplementary Figure 14. HCSVs had been efficiently delivered to the tumor through I.T. infusion with MRI imaging. The localized HCSVs and magnetic field treated only tumor regions showing increased positive TUNEL tissues. Hematoxylin and eosin (H&E) histology images of the major organs (i.e., liver, spleen, lung, and kidneys) after the treatment demonstrated that there is no noticeable tissue damage in the collected organs (Supplementary Figure 14). We add this information in our revision, as shown below.*

(line 183 in Page 7)

Terminal deoxynucleotidyl transferase dUTP nick end labeling (TUNEL)-stained tumor slices, which were collected 24 h post 1st local treatment, were shown in **Figure 3f**. The TUNEL stain showed a robust conclusion on the therapeutic effect of ferroptosis. At 14 day-post treatment, the major organs (i.e., liver, spleen, lung, and kidneys), which might be affected by nanomaterials -based therapy,³⁰ were sliced and stained with Hematoxylin and eosin (H&E) for histology analysis. The results demonstrated that there is no noticeable tissue damage in the collected organs (**Supplementary Figure 14**). In addition, we tried to treat the animal with the combination of MF and HCSVs through i.v. of HCSVs. A dramatic growth curve was found and there was no sign of therapeutic effect was proved (**Supplementary Figure 15**). This might be due to low tumor accumulation efficiency of HCSVs, which is the main challenge in nanomedicine.³¹ The scope in work was to prove an MRI visible and MF controllable method by utilizing HCSVs with a special core-shell design through I.T., which a local therapy modality was developed to treat cancer as the clinical method in Radiology.^{32, 33}

(line 333 in Page 11)

Terminal deoxynucleotidyl transferase dUTP nick end labeling (TUNEL) stain. To evaluate the antitumor therapeutic efficacy, the tumor tissues were harvested 24 h after various treatment described in vivo therapy. Hematoxylin and eosin (H&E) stain. To analyze the toxicity toward the normal organs, liver, spleen, lung, and kidney were collected at 14 days post treatment. The harvested tissues were fixed with 10% neutral formalin solution and then submitted to Mouse Histology and Phenotyping Laboratory (MHPL) core for TUNEL staining with 5 μ m slice thickness.

(line 185 in Page 7)

At 14-day post treatment, the major organs (i.e., liver, spleen, lung, and kidneys), which might be affected by nanomaterials-based therapy,³⁰ were sliced and stained with Hematoxylin and eosin (H&E) for histology analysis.

The results demonstrated that there is no noticeable tissue damage in the collected organs (**Supplementary Figure 14**).

Figure 3. Combination therapy-mediated antitumor effect in a TRAMP -C1 tumor model. **(a)** in vivo treatment timeline. **(b)** tumor growth curves of the TRAMP-C1. **(c)** the record of mouse weight. **(d)** tumor weight at the end time point. **(e)** the typical *ex vivo* photo of dissected tumors from various treatment. (n=6) **(f)** TUNEL-stained tumor slices (scale bar: 2.5 mm). Tumor tissues collected from different groups at 24 h after various treatments.

Supplementary Figure 14. H&E-stained organ (liver, spleen, lung, and kidneys) slices collected 14 days after various treatment (control: mice without treatment, HCSVs+MF: mice treated with combination of MF and HCSVs as described *in vivo* treatment). The results demonstrated that there was no noticeable tissue damage in the collected organs.

Comment 6: Linear relationship should be presented in Figure 2 a, as well as the slope and R square.

Response: *We appreciate the reviewer's comment. As reviewer commented, Figure 2a is updated with linear relationship with slope and R square values. Please see below.*

(line 153 in Page 5) *updated figure shown as below*

Figure 2. (a) *In vitro* $R2^*$ value of IONCs treated with/without AA, R square are 0.9963 and 0.8287 for IONCs (black curve) and IONCs+AA (red curve), respectively (b) *In vivo* $R2^*$ change after intra-tumoral injection of HCSVs treated with/without MF. R square are 0.9520 and 0.9741 for red curve (with MF) and blue curve (without MF), respectively. (n=3) (c) Related $R2^*$ mapping after intra-tumoral injection of HCSVs treated with/without MF. (red circle indicated tumor region).

Comment 7: Data were obtained at only two time points in Figure 2 b. It is too limited. At least 4 time points should be presented to draw a conclusion. MR images of the tumor-bearing mice should be presented to show the MRI signal change at tumor site after injection of the HCSVs.

Response: *We appreciate the reviewer's comment. Here we added more time points including Pre, Post, Post 4h and Post 12h and updated Figure 2b. At the same time, $R2^*$ mapping images (Figure 2c) were added to show clear MR signal changes.*

(line 153 in Page 5) *updated figure shown as below*

Figure 2. (a) *In vitro* R2* value of IONCs treated with/without AA, R square are 0.9963 and 0.8287 for IONCs (black curve) and IONCs+AA (red curve), respectively (b) *In vivo* R2* change after intra-tumoral injection of HCSVs treated with/without MF. R square are 0.9520 and 0.9741 for red curve (with MF) and blue curve (without MF), respectively. (n=3) (c) Related R2* mapping after intra-tumoral injection of HCSVs treated with/without MF. (red circle indicated tumor region).

Comment 8: For the animal studies, intravenous injection should be used instead of intra-tumoral injection.

Response: Thank you very much for your comment. IV injection of nanomedicine is an ideal approach and has shown promising results for the potential therapeutic applications. However, limited targeting efficiency from recent report negatively affects the translation of nanomedicine to clinical applications (Syed, A. M., et al. *Nat Biomed Eng* 2017, 1,629.). Hence, current future cancer nanomedicine strongly requires more localized approaches considering the tumor heterogeneity. As demonstrated in interventional oncology area (Kim, D., J. *Imaging* 2018, 4, 18.; Guan, J. Y., et al. *J Vasc Interv Radiol* 2011, 22, 1216.), intra-tumoral injection along with appropriate image guidance is one of the promising approaches to overcome current challenge in nanomedicine. Thus, the I.T. infusion of HCSVs with MRI would be feasible approach to translate the novel nanomedicine strategy. As shown in our results, our local delivered HCSVs eventually contributed to significant tumor growth inhibition. As reviewer mentioned, I.V. injection was tried to compare the potential therapeutic effect. We treated the animal with the combination of MF and HCSVs through i.v.. A rapid tumor growth curve was found and there was no sign of therapeutic effect was proved, as shown in **Supplementary Figure 14**. This might be due to low tumor accumulation efficiency of HCSVs, which is the main challenge in nanomedicine. (Syed, A. M., et al. *Nat Biomed Eng* 2017, 1,629.).

(line 188 in Page 7)

In addition, we tried to treat the animal with the combination of MF and HCSVs through i.v. of HCSVs. A dramatic growth curve was found and there was no sign of therapeutic effect was proved (**Supplementary Figure 15**). This might be due to low tumor accumulation efficiency of HCSVs, which is the main challenge in nanomedicine.³¹ The scope in work was to prove an MRI visible and MF controllable method by utilizing HCSVs with a special core-shell design through I.T., which a local therapy modality was developed to treat cancer as the clinical method in Radiology.^{32, 33}

(line 329 in Page 11)

Supplementary Figure 15. Tumor growth curves of the TRAMP-C1 bearing mice treated with intravenous injection (I.V.) of HCSVs (50 μ L, 10 mg/Kg) following with MF treatment for 14 days post I.V.(n=5). Those mice were repeatedly treated three times at 0-day, 3-day, and 6-day time point.

Thank you very much for your valuable and insightful comments. We are sure that these comments help improving the quality of the manuscript.

Reviewer #2

General comments:

The manuscript, “Exogenous Magnetic-field Boosted Ferroptosis-mediated Immune Response and Responsive Magnetic Resonance Imaging Using Hybrid Core-shell Vesicles for Cancer Therapy” investigates an approach to enhance the safety and efficacy of ferroptosis inducing nanoparticles as a cancer therapy. Using biosafe materials for iron oxide nanocubes, the authors synthesized hybrid core-shell vesicles (HCSVs). Notably, the Fenton reaction and subsequent ferroptosis induced by these nanoparticles can be triggered by an exogenous circularly polarized magnetic field (MF) – thus allowing for improved control over the release of the therapeutic effect. The authors show that the combination of HCSVs with MF results in strong suppression of tumor growth, however there are some concerns with the robustness of the in vivo experiments as well as the conclusions drawn from immunological mechanisms involved. Points to address include:

Response: *We really appreciate your all comments. Please see our response with additional data, and revised sentences marked with yellow background.*

Comment 1: The main concern with this manuscript is the claim that upregulation of calreticulin (CRT) is the key mechanism responsible for inducing/increasing immune response. It is shown to be correlated with immune cell activation but a causative effect is not validated. As such, it is recommended that an additional group is tested such that a blocking antibody against CRT is incorporated to verify that it is indeed a direct cause of improved DC and CTL responses reported.

Response: *We appreciate for the reviewer’s valuable comment. As reviewer mentioned, increasing evidence have well proved that the translocation of CRT from the endoplasmic reticulum to the cell surface dictates the immunogenicity of cancer cell death. (Obeid, M., et al. Cancer Res 2007, 67, 7941.). To validate this further in our study, an experiment was performed with an additional group with anti-CRT. All splenocytes from the variously treated or untreated mice were cultured with Tramp-C1 cells which treated with HCSVs+MF for 48 hours. As compared to the control group, the combination group of HCSVs+MF remarkably promoted the proliferation of splenocytes by near 2-fold (Supplementary Figure 16). After blockade of the exposure CRT on the treated Tramp-C1 cells by anti-CRT at 4 °C for 1 h, the optical density at 450 nm was significantly decreased to be 1.16±0.24. Taken together, CRT exposure contributed to increase immune response in the combination therapy. This data is added in Supplementary Figure 16, as shown below.*

(line 199 in Page 7)

Herein, a splenocytes proliferation assay was performed to confirm if CRT exposure contributed to increase immune response in the combination therapy. As shown in **Supplementary Figure 16**, the combination group of HCSVs+MF remarkably promoted the proliferation of splenocytes by near 2 folds of control group. After blockade of the exposure CRT on the treated TRAMP-C1 cells by anti-CRT at 4 °C for 1 h, the optical density at 450 nm was significantly decreased to be 1.16±0.24. Taken together, CRT exposure contributed to increase immune response in the combination therapy.

(line 345 in Page 11)

Splenocytes proliferation assay. Splenocytes obtained from treated mice were re-stimulated by antigens, and the cell proliferation was measured by a CCK-8 kit assay following previous report.³⁵ TRAMP-C1 cells in 6-well plate was treated with combination procedure of MF and HCSVs described in cell viability assay for 4 h, then was collected with trypsin. Then splenocytes (5×10^5 per well) were seeded in the 96-well plate and retreated with the treated TRAMP-C1 cells (5×10^4 per well) for 48 h. Splenocytes proliferation was measured using CCK-8 kit analysis. OD values at 450 nm were recorded by a microplate reader.

Supplementary Figure 16. Splenocytes proliferation assay. Splenocytes obtained from treated mice were re-stimulated by antigens, and the cell proliferation was measured by a CCK-8 kit assay.

Other concerns include:

Comment 2: Clarification of number of mice used for each group in the in vivo experiments

Response: *We appreciate reviewer's careful comment. Six mice had been used for each group. This information is added in the revised manuscript.*

(line 159 in Page 6) *Caption of Figure 3*

(line 220 in Page 9) *Caption of Figure 4*

(line 329 and line 331 in Page 10) *"In vivo therapy" section.*

Comment 3: Clarification of timeline for tumor model – were TRAMP-C1 tumor cells and Treatment 1 both injected on same day? If so, treatment of mice with a more developed tumor would provide more confidence in the efficacy of the approach.

Response: *We appreciate reviewer's careful comment. We performed tumor implantation of TRAMP-C1 cells 14 days before carried out of ISt treatment towards animals. We updated Figure 3a as shown below.*

(line 158 in Page 6)

Figure 3. Combination therapy-mediated antitumor effect in a TRAMP -C1 tumor model. (a) in vivo treatment timeline. (b) tumor growth curves of the TRAMP-C1. (c) the record of mouse weight. (d) tumor weight at the end time point. (e) the typical *ex vivo* photo of dissected tumors from various treatment. (n=6) (f) TUNEL-stained tumor slices (scale bar: 2.5 mm). Tumor tissues collected from different groups at 24 h after various treatments.

Comment 4: The percentage values reported in the bar graphs of Figure 4 should specify whether they are defined in relation to all cells in tumor/lymph nodes or just relative to the T cell populations

Response: *We appreciate reviewer's careful comment. The percentage values in the graphs of Figure 4 were defined in relation to T cell populations. Please see the updated Figure 4.*

(line 219 in Page 8)

The percentage values in the graphs were defined in relation to T cell populations.

Figure 4. Immune responses after treatment to TRAMP-C1 tumor-bearing mice. **(a, b)** Flow cytometric analysis images and the statistical data **(b)** for *in vivo* DC maturation. Cells in the tumor-draining lymph nodes were collected after various treatments for the assessment by flow cytometry after staining with CD11c, CD80, and CD86. **(c, d)** Representative flow cytometric analysis images showing CD8⁺ T cells (CD3⁺ CD4⁻ CD8⁺) from different groups of mice. Proportions of tumor infiltrating CD8⁺ killer T cells in the tumor (up line) and draining lymph nodes (down line) **(c)** and the statistical data in the tumor **(d)** and in lymph node **(e)** among all cancer cells (n=6). The percentage values in the graphs were defined in relation to T cell populations.

Comment 5: The flow cytometry graphs in Figure 4 should include percentages of the gated populations to corroborate the values shown in bar graphs.

Response: *We appreciate reviewer's careful comment. The percentage values are added in the updated graphs of Figure 4, as shown in below.*
(line 219 in Page 8)

Figure 4. Immune responses after treatment to TRAMP-C1 tumor-bearing mice. **(a, b)** Flow cytometric analysis images and the statistical data **(b)** for *in vivo* DC maturation. Cells in the tumor-draining lymph nodes were collected after various treatments for the assessment by flow cytometry after staining with CD11c, CD80, and CD86. **(c, d)** Representative flow cytometric analysis images showing CD8⁺ T cells (CD3⁺ CD4⁻ CD8⁺) from different groups of mice. Proportions of tumor infiltrating CD8⁺ killer T cells in the tumor (up line) and draining lymph nodes (down line) **(c)** and the statistic data in the tumor **(d)** and in lymph node **(e)** among all cancer cells (n=6). The percentage values in the graphs were defined in relation to T cell populations.

Thank you very much for your valuable and helpful comments. We are sure that the reviewer's critical and insightful perspectives improved the quality of the manuscript

Reviewer #3

General comments:

In the paper “Exogenous Magnetic-field Boosted Ferroptosis-mediated Immune Response and Responsive Magnetic Resonance Imaging Using Hybrid Core-shell Vesicles for Cancer Therapy” Yu et al present a Hybrid Core-shell vesicle (HCSV) vesicle able to release ascorbic acid (AA) in the presence of a magnetic field that subsequently leads to reduction of iron and its release. My comments to the paper restrain to the ferroptosis part as I cannot judge the novelty and critical aspect of the HCSV synthesis and usage. Pointing this out there are several issues with the authors conclusion when they point out that ferroptosis is been triggered and this is the cause for the immune response observed; and additional experiments are required in order to substantiate their claims.

Response: *We really appreciate your all comments related with ferroptosis. Please see our response with additional data, and revised sentences marked with yellow background.*

Comment 1: The authors claim ferroptosis is responsible for cell only based on a mild rescue effect observed with Ferrostatin 1. More evidence should be provided, do increase content of PUFA sensitize further? Does GPX4 OE expression is protective? This should be a minimal set to implicate lipid peroxidation in the phenomenon observed.

Response: *We much appreciate reviewer’s insightful comments. Ferroptosis is a non-traditional form of cell death resulting from iron-dependent lipid peroxide accumulation. Studies have demonstrated that ferroptosis is associated with a variety of different types of cancer (Mou, Y., et al. Li, B. J Hematol Oncol 2019, 12, 34.; Dixon, S. J., et al. Cell 2012, 149, 1060.; Chen, G., et al. Oncol Lett 2020, 19,579.) There are two main pathways, which are iron metabolism and reactive oxygen species metabolism, respectively, involved in ferroptosis (Song, Y., et al. Oncol Lett 2019, 18, 2159). Our MF controllable therapeutic system, HCSVs, was used to supply ferrous ions to cancer cells. Thus, this strategy was directly related to increased iron metabolism, which led to radical accumulation and result in lipid hydroperoxides. Ferrostatin-1 is an active radical-trapping antioxidant that traps peroxy radicals, and thus serves as a potential inhibitor of ferroptosis (Miotto, G., et al. Redox Biol 2020, 28.). Furthermore, according to our result in **Figure 1d**, the significant difference of cell viability was found in Ferrostatin-1 treated group as compared to HCSVs+MF treatment. It was confirmed: first, our therapy strategy directly affected iron metabolism and resulted in lipid hydroperoxides, second, Ferrostatin-1, a ferroptosis inhibitor, significantly protected cells from the iron-metabolism related cell damage. For the reactive oxygen species metabolism, accumulated lipid hydroperoxides in cells are normally detoxified by a GPX family member called glutathione peroxidase 4 (GPX4) that uses GSH to convert lipid hydroperoxides to lipid alcohols and thus suppresses ferroptosis (Koppula, P., et al. Cancer Commun 2018, 38.) Newly added results in **Supplementary Figure 13** showed the combination treatment with MF and HCSVs downregulated GPX4. On the other hand, Ferrostatin-1 treatment recovered GPX4 level inside cells. We assumed that the continuous depletion of GPX4 by HCSVs-boosted lipid hydroperoxides contributed to the downregulation of GPX4, while the protection of Ferrostatin-1 reduced the consumption of GPX4 that increased the GPX4 level inside cells. Thus, considering the straightforward connection of HCSVs-boosted iron metabolism and hydroperoxide-related cell death, our results confirmed the potential of MF controllable ferroptosis-mediated cell-killing strategy.*

(line 134 in Page 5)

There are two main pathways, which are iron metabolism and reactive oxygen species metabolism, respectively, involved in ferroptosis.²⁸ Herein, our MF controllable therapeutic system, HCSVs, was directly used to supply ferrous ions to cancer cells. This strategy was directly related to boost iron metabolism, which led to radical accumulation and result in lipid hydroperoxides. For the reactive oxygen species metabolism, accumulated lipid hydroperoxides in cells are normally detoxified by a GPX family member called glutathione peroxidase 4 (GPX4) that uses glutathione (GSH) to convert lipid hydroperoxides to lipid alcohols and thus represses

ferroptosis.²⁷ As shown in **Supplementary Figure 13**, the combination treatment of MF and HCSVs downregulated GPX4. On the other hand, Ferrostatin-1 treatment recovered GPX4 level inside cells. We assumed that the continuous depletion of GPX4 by HCSVs-boosted lipid hydroperoxides contributed to the downregulation of GPX4, while the protection of Ferrostatin-1 reduced the depletion of GPX4 that finally increased the GPX4 level inside cells.

(line 312 in Page 11)

Western Blotting Analysis. TRAMP-C1 cells were seeded in the 8 cm culture dishes and cultured at 5% CO₂, 37 °C overnight. The cells were treated with various procedures, 400 µg/mL of HCSVs +MF+ 200 nM of Ferrostatin-1, 400 µg/mL of HCSVs, 400 µg/mL of HCSVs +MF, 600 µg/mL of HCSVs +MF, 800 µg/mL of HCSVs +MF, respectively, for 12 h. No treatment of TRAMP-C1 was used as control. The cell lysates were collected and analyzed by Western blotting according to the manufacturer's instructions.

Supplementary Figure 13. (a) Western blot analysis of Tramp-C1 cells treated with various methods for 12 h. (b) Quantification of relative bands of western blotting. Lane 1 to Lane 6 were treated with HCSV(400 µg)+MF+ Ferrostatin-1, Control, HCSVs (400 µg), HCSVs (400 µg)+MF, HCSVs (600 µg)+MF, HCSVs (800 µg)+MF, respectively. HCSVs (400 µg): 400 µg/mL of HCSVs, HCSVs (600 µg): 600 µg/mL of HCSVs, HCSVs (800 µg): 800 µg/mL of HCSVs.

Comment 2: The TRAMP-C1 cancer line deserve some characterization in the context of how it responds to standard ferroptosis inducing agents as well as providing expression data on key players such as GPX4, SLC7A11, γ-GCS, ACSL4 etc.

Response: *We much appreciate reviewer's insightful comments. As mentioned in previous comment, for the ROS metabolism, GPX4 suppressing ferroptosis is one of key marker to prove ferroptosis. Here we tested further the ferroptosis effect of the proposed MF and HCSVs with Western blot analysis of GPX4. Western blot analysis of Tramp-C1 cells treated with various methods was added to Supplementary Figure 13. HCSVs-boosted iron metabolism strategy was used to inhibit tumor growth. No small molecular agents that are related to protein targeting or structure-activity were used in this work. Thus, based on the therapy logics, we tested GPX4, a protein regarding ROS metabolism. As shown and discussed in response to question 1, our data with TRAMP-C1 cancer cells indicated that the consumption of GPX4 by HCSVs-boosted lipid hydroperoxides contributed to the downregulation of GPX4, while the protection of Ferrostatin-1 reduced the consumption of GPX4 that increased the GPX4 level inside cells.*

(line 134 in Page 5)

There are two main pathways, which are iron metabolism and reactive oxygen species metabolism, respectively, involved in ferroptosis.²⁸ Herein, our MF controllable therapeutic system, HCSVs, was directly used to supply ferrous ions to cancer cells. This strategy was directly related to boost iron metabolism, which led to radical accumulation and result in lipid hydroperoxides. For the reactive oxygen species metabolism, accumulated lipid hydroperoxides in cells are normally detoxified by a GPX family member called glutathione peroxidase 4 (GPX4) that uses glutathione (GSH) to convert lipid hydroperoxides to lipid alcohols and thus represses ferroptosis.²⁷ As shown in **Supplementary Figure 13**, the combination treatment of MF and HCSVs downregulated GPX4. And Ferrostatin-1 treatment recovered GPX4 level inside cells. We assumed that the continuous depletion of GPX4 by HCSVs-boosted lipid hydroperoxides contributed to the downregulation of GPX4, while the protection of Ferrostatin-1 reduced the depletion of GPX4 that finally increased the GPX4 level inside cells.

(line 312 in Page 11)

Western Blot Analysis. TRAMP-C1 cells were seeded in the 8 cm culture dishes and cultured at 5% CO₂, 37 °C overnight. The cells were treated with various procedures, 400 µg/mL of HCSVs +MF+ 200 nM of Ferrostatin-1, 400 µg/mL of HCSVs, 400 µg/mL of HCSVs +MF, 600 µg/mL of HCSVs +MF, 800 µg/mL of HCSVs +MF, respectively, for 12 h. No treatment of TRAMP-C1 was used as control. The cell lysates were collected and analyzed by Western blotting according to the manufacturer's instructions.

Supplementary Figure 13. (a) Western blot analysis of Tramp-C1 cells treated with various methods for 12 h. (b) Quantification of relative bands of western blotting. Lane 1 to Lane 6 were treated with HCSV(400 µg)+MF+ Ferrostatin-1, Control, HCSVs (400 µg), HCSVs (400 µg)+MF, HCSVs (600 µg)+MF, HCSVs (800 µg)+MF, respectively. HCSVs (400 µg): 400 µg/mL of HCSVs, HCSVs (600 µg): 600 µg/mL of HCSVs, HCSVs (800 µg): 800 µg/mL of HCSVs.

Comment 3: The in vivo effect is indeed dramatic and could well be that immune cells are further boosting the antitumor effect. If indeed ferroptosis is triggering this process, how ferroptosis inhibitors would modulate this response in vivo – Perhaps even more important would be to check how tumors respond in different genetic background such as Batf3 KO mice that fail to mount an innate immune response.

Response: Thank you very much for reviewer's suggestion. The increase of intracellular iron ions and follow-up Fenton reaction, which elevates ROS levels, leads to ferroptosis cell death by irresistible lipid peroxidation. The intense membrane lipid peroxidation and consequential loss of selective permeability of the plasma membrane are characterized in ferroptosis. (Yagoda, N., et al. Nature 2007, 447, 864.); Yang, W.S., et al. Cell 2014, 156, 317.) Generally, ferroptosis-induced cell death has been proved to be effective in killing cancer cells through ROS accumulation in cells. (Kroemer, G., et al. Cell Death and Differentiation 2009, 16, 3; Galluzzi, L., et al. Cell Death and Differentiation 2012, 19, 107.) Moreover, recent studies proved that the oxidative stress-inducing ferroptosis also upregulates translocation of calreticulin (CRT) expression on the surface of tumor cells. The phagocytic "eat me" CRT signal induces robust antitumor immune responses by eliciting

phagocytosis of tumor-associated antigens. Herein, our therapy strategy induced a stronger ROS accumulation (Supplementary Figure 10) and lipid peroxidation (Supplementary Figure 11 and Supplementary Figure 12), which led to CRT exposure (Figure 1e and 1f) and finally resulted in CRT-based immune response (Supplementary Figure 16). According to current data, our results well confirmed the hypothesis in this work. Suggested Batf3 KO mice would provide useful advanced information to prove further ferroptosis mechanism. However, here we focus more on the proof-of-concept to open a new potential application of ferroptosis inducing agents and ferroptosis mediated immune response. The suggestion will be added to our future study plan. Thanks again for your helpful suggestion.

Comment 4: Methodologies used to monitor “ROS” and LPO are antiquated and several caveats for these methodologies have been extensively reported in the literature. More importantly the kit used to measure LPO is not measuring MDA as stated, without chromatographic separation this cannot be stated.

Response: *We appreciate the reviewers' insightful and helpful comments. We fully agree with reviewer's opinion. In this revision, a lipophilic fluorescent dye 4,4-difluoro-5-(4-phenyl-1,3-butadienyl)-4-bora-3a,4a-diaza-s-indacene-3-undecanoic acid (C11-BODIPY581/591) to probe oxyl-radical induced lipid oxidation inside cells was used to measure LPO. C11-BODIPY581/591 is a fatty acid analog with specific fluorescent properties (Drummen, G. P., et al. J. A. Free Radic Biol Med 2002, 33, 473.). When excited with blue light at 488 nm wavelength, the molecule has a constitutive fluorescence emission with a maximum at 595 nm. Following oxyl-radical induced oxidation, the fluorescence emission shifts to shorter wavelengths with a maximum emission at 520 nm. Due to its lipophilic properties, the molecules easily enter the lipid bilayer and once inside the cellular membrane are subject to oxidation by oxyl-radicals together with the endogenous fatty acids (Cheloni, G., et al. Cytometry A 2013, 83, 952.). As shown in Supplementary Figure 11, the combination of MF and HCSVs triggered LPO.*

(line 126 in Page 4)

C11-BODIPY581/59, an oxidation-sensitive fluorescent lipid peroxidation probe, have been widely used as a detector for ferroptosis.²⁶ The shift of red to green fluorescence indicated the lipid peroxidation. As the transferrin receptor on the cell membranes will affect the cellular uptake of ferrous and ferric ions,²⁷ herein, mesoporous silica nanoparticles (MSN) was used as a carrier for both ferrous and ferric ions. As shown in **Supplementary Figure 11**, cells treated with ferrous-MSN showed a higher green fluorescence as compared to ferric-MSN treated cells which indicated higher level of LPO. This might be due to a stronger Fenton reaction induced by ferrous ions as compared to ferric ions.²³ A higher ratio of green to red signal also found in the combination of HCSVs and MF

(line 263 in Page 9)

Synthesis of mesoporous silica nanoparticles (MSN) and iron ions loaded MSN. MSN were synthesized by an oil-water biphasic reaction approach by following a reported procedure.³⁵ In details, 20 mL of CTAC solution (25 wt %) and 0.01 g of TEA were mixed with 30 mL of water and gently stirred at 50 °C for 1 h. Then 15 mL of TEOS in cyclohexane (5% v/v) was slowly added to the above solution and kept at 50 °C for another 18 h. Finally, MSN were collected by centrifugation at 14000g for 15 min. The precipitates were washed four times (24 h/time) with 1% (wt %) NaCl/methanol solution to remove CTAC. The structure of MSN was confirmed by TEM. Iron ions was dissolved in the water. The amount of water is 1.5 times the weight of MSN. Then the dry MSN powder was mixed with iron ions aqueous solution for 2 h. Above mixture of MSN and iron ions was dry under Argon flow overnight. Finally, iron ions loaded MSN was washed with ethanol and was kept in dry powder for next application. The iron weight ratio in MSN were 17.9% and 18.4% for ferrous and ferric ion, respectively.

(line 308 in Page 9)

The cells were stained with C11- BODIPY581/591 (2 μ M) and incubated for 30 min. The cells were treated with various materials by following the procedure described in cell viability test. A microplate detection Cytation 3 (BioTek Instruments, Inc.) was used to detect the fluorescence intensity at 530 nm (green) and 591 nm (red).

Supplementary Figure 11. Lipid peroxide stained with fluorescent C11-BODIPY581/591 in cells after coincubation with ferrous/ferric ions loaded mesoporous silica nanoparticles (MSN). (a) TEM images of MSN. (b) Fluorescence emission intensity of C11-BODIPY581/591 at 530 nm (green) and 591 nm (red) in cells after coincubated with ferrous-MSN (Ferrous-Green or Ferrous-Red) or ferric-MSN (Ferric-Green or Ferric-Red). (c) Ratio of green:red fluorescence intensity in cells after coincubated with ferrous-MSN or ferric-MSN. (d) Ratio of green:red fluorescence intensity in cells after coincubated with HCSVs following with/without MF treatment.

Thank you very much for appropriate and valuable comments. We are sure that these comments improved significantly the quality of the manuscript.

REVIEWERS' COMMENTS:

Reviewer#1:

The authors have adequately addressed the concerns raised by this reviewer.

Reviewer#2:

The authors have adequately addressed previous concerns by clarifying experimental procedures and performing additional assays that more strongly support the proposed mechanisms involved. Furthermore, the work done in response to other reviewer comments similarly added robustness to the manuscript that strengthened the validity of the results presented in the original work. Having made these changes, I now recommend the updated manuscript for acceptance for publication.

Reviewer#3:

no more comments

Reviewer#1:

The authors have adequately addressed the concerns raised by this reviewer.

Response: *We really appreciate your all positive critics and comments.*

Reviewer#2:

The authors have adequately addressed previous concerns by clarifying experimental procedures and performing additional assays that more strongly support the proposed mechanisms involved. Furthermore, the work done in response to other reviewer comments similarly added robustness to the manuscript that strengthened the validity of the results presented in the original work. Having made these changes, I now recommend the updated manuscript for acceptance for publication.

Response: *We express our best gratitude for your kind comments and guidance.*

Reviewer#3:

no more comments

Response: *We appreciate the reviewer's time and comment to help us to improve scientific rigor in our work. Herein, we change the phase of "Ferroptosis" to "Ferroptosis-like cell death" in our manuscript.*